# Investigating the test-retest reliability of Illinois Agility Test for wheelchair users

**Zohreh Salimi**[ORCID]¤, **Martin William Ferguson-Pell***

Department of Rehabilitation Science, Faculty of Rehabilitation Medicine, University of Alberta, Edmonton, Alberta, Canada

¤ Current address: Autism Spectrum Disorders Research Center, Hamedan University of Medical Sciences, Hamedan, Iran.
* martin.ferguson-pell@ualberta.ca

## Abstract

The Illinois Agility Test (IAT) is a standard agility course used to assess and train able-bodied athletes as well as wheelchair-sport athletes and rehabilitation of wheelchair users. It has been shown to be a reliable and valid tool to assess the able-bodied population, but the reliability of this test for assessing wheelchair propulsion has never been shown. The purpose of this study is to investigate the test-retest reliability of IAT to assess wheelchair propulsion. In this paper, the test-retest reliability of using IAT for wheelchair users is found for peak and average velocity, acceleration, tangential and total force of the push, each for the left and the right wheel. Each of these variables was found for thirty-two decisive points throughout the IAT path. The Intra-class Correlation Coefficient (ICC) was found to be very strong for 15 out of 16 variables. The average ICC was 89% and the average 95% confidence interval was [44% 96%]. In addition, thirty-seven other significant propulsion parameters were found that are clinically important, such as the number of pushes participants take to go around cones on the right, relative to turning around the cones on the left. Also, all thirty-seven variables were compared between the two sessions using four separate MANOVAs; the results showed no significant difference between IAT performed in the two sessions which were at least one week apart. This, in turn, backs the reliability of IAT for wheelchair users. These results are sufficient evidence to show that IAT is a reliable tool to test wheelchair agility for fifteen variables tested for non-wheelchair users. Since experienced wheelchair users are much more consistent in wheelchair propulsion compared to non-wheelchair-users, the results of this study show that IAT can be used as a reliable tool to assess and train wheelchair users, both for clinical and athletic applications.

## Introduction

Wheelchair propulsion requires an intense effort that wheelchair users repeatedly undergo every day. The intensity of effort is firstly because the efficiency of wheelchair propulsion is almost half the efficiency of walking (~14% [1] vs 26 to 27% [2]), and secondly because

**Data Availability Statement:** All data files are available from the BioStudies Repository database (accession number S-BSST373).

**Funding:** The authors received no specific funding for this work. Canadian Foundation for Innovation

(CFI 30852 Leading Edge Fund 2012) infrastructure grant provided the capital equipment for the project.

**Competing interests:** The authors have declared that no competing interests exist.

wheelchair users use their arms instead of their legs to ambulate which are not evolved for that purpose [3]. The shoulder joint is therefore vulnerable when wheelchairs are used as a primary method of ambulation. In fact, about 20% of wheelchair users develop secondary injuries in their upper extremities around five years after starting to use a wheelchair [4]. This percentage increases to 46% about 20 years after the injury [4, 5]. This mainly happens because of the excessive forces exerted on the structures around the shoulder during wheelchair propulsion over rough terrains and also manoeuvring [6].

Manoeuvring is performed extensively for indoor wheelchair use [7] and is responsible for a greater portion of the excessive loading on upper extremities [8] and yet it has seldom been studied in the literature. Having standard and reliable tests for measuring wheelchair manoeuvres will help to develop a more sophisticated understanding of the biomechanics of manoeuvring.

The Illinois Agility Test (IAT) is a standard agility test that has been used for both training and assessment of able-bodied athletes for many years [9, 10]. It was introduced for measuring multidirectional agility for different sports [9, 11]. For instance, IAT has been used to test the effect of strengthening exercises and also to test athletes' agility in soccer [12]. However, some researchers have used IAT for wheelchair athletes as well. William [13] tried to find physiologic determinants to assess the training of wheelchair basketball players [14] and Usma-Alvarez, et al. have considered developing some outcomes for IAT that are suitable in assessing wheelchair rugby.

We could not find any publications that have shown the reliability of an agility test for wheelchair users, although there are a few papers that have investigated the reliability of different agility tests for the able-bodied population. Namely, *within-day* reliability of IAT for 66 able-bodied semi-professional rugby players has been shown to be ICC = 86% [15]. In another study, the reliability of IAT was found on a sample of 97 able-bodied young military service men: ICC = 99% inter-rater reliability and ICC = 68% test-retest reliability [9]. Also, an ICC of 96% was reported elsewhere for IAT for a sample of 89 able-bodied sportsmen from football, handball, and rugby players when repeating the test on different days [11].

Williams [14] has investigated the construct validity of three different agility tests, including IAT, for athlete wheelchair basketball players. They compared the timed performance of a group of elite athletes to a group of competitive players and concluded the test was valid as there was a significant difference in the time to finish the test. They also found excellent test-retest reliability with correlations above 0.9 between 2 trials performed in one session. However, they only measured the time to finish the test and not other important biomechanical factors. This is important because when using an assessment tool for measuring an outcome we need to be confident that this tool will create similar results if used sometime later [16].

An important factor when testing agility is how fast the participant could finish the agility path. This is the reason that the primary output of all agility tests is the time needed to finish the task. However, a good agility test should be designed in a way to measure the acceleration of the participant, as changes in the direction, which are the main parts of agility tests, are more correlated with acceleration than speed [17]. Although completion time is not a great concern when studying wheelchair maneuvering, the acceleration assessment makes agility tests suitable for the purpose of this study, as it is closely related to maneuverability for wheelchair users. The purpose of this study was to investigate the test-retest reliability of IAT in order to introduce a reliable standard agility test that can be used not only for studying wheelchair maneuvering but also for training athlete wheelchair users.

## Methods

### Sample size calculation

To calculate a sample size that provides an acceptable power for finding the reliability of IAT, we used Eq 1 that is introduced by Shoukri et al. [18]:

$$k = 1.5 + \left[ \frac{z_\alpha + z_\beta}{\mu(\rho_0) - \mu(\rho_1)} \right]^2 \qquad (1)$$

Where K is sample size, $z_\alpha$ is the value of the standard normal distribution corresponding to the level of significance ($\alpha$) of this study, $z_\beta$ is the value of the standard normal distribution corresponding to the power of this study (1-$\beta$), $\mu$ is mean of the distribution of reliability, $\rho_0$ is the minimum reliability coefficient we can tolerate, and $\rho_1$ is the estimated reliability coefficient of the measure. This equation calculates the needed sample size when the number of repetitions of the test per subject is two.

$\mu(\rho)$ is calculated using Eq 2:

$$\mu(\rho) = \frac{1}{2} \ln \left( \frac{1 + \rho}{1 - \rho} \right) \qquad (2)$$

The minimal acceptable reliability was set as 0.6, as it is the minimum acceptable reliability coefficient for many clinical investigations [18]. So, for having the significance level of $\alpha = 0.05$, $\beta = 0.2$ (power of 80%), $\rho_0 = 0.6$ and $\rho = 0.9$, Eq 1 would give 11.63. Thus, the sample size needed for this study is obtained as 11 persons.

### Subjects

Eleven healthy able-bodied subjects volunteered to participate in the study (6 women and 5 men: (mean ± SD) age = 27.9 ± 4.74 years, weight = 63.5 ± 10.96 Kg, and height = 1.69 ± 0.1 m). The ethics application for this study was approved by University of Alberta Research Ethics Board, Project Name "Can simulation be used to accurately represent real-life wheelchair propulsion?", No. Pro00003315 _AME5, Date: 9/25/2015. Subjects signed a written informed consent form, a video release form, as well as a "PAR_Q and You" (Physical Activity Readiness Questionnaire) prior to participating. participants were excluded if they put "yes" for any of the questions in "PAR_Q and You" or if they had a musculoskeletal injury that affects normal wheelchair use, exercise-induced asthma, or heart disease. Participants were a convenience sample recruited from graduate or visiting students of the corresponding author's university. One participant was Korean, two were Hispanic, 5 were White, and 3 were Persian.

**Experiment content.** Subject participation was according to an experimental protocol that was approved by the Human Research Ethics Board of the University of Alberta (ID: Pro00003315_AME6). Each subject had two sessions of testing that were at least one week apart (mean± SD = 10±4.1 days); In the first session, they performed four trials and at the second session they performed eight trials of IAT (Fig 1). Subjects were given enough time between the trials (at least three minutes) in both sessions to rest. The next trial did not start until the participants declared they had had sufficient rest and their heart rate returned to their resting heart rate.

IAT is an obstacle course consisting of 8 cones that are positioned as shown in Fig 2. Participants were asked to run the path (shown in Fig 2) as quickly as they could. IAT was originally designed to start from the prone position; the person performing the test was to lay down in the prone position and at the beginning of IAT he/she would quickly stand up and start running [9]. However, IAT sometimes has been used without the prone start [15].

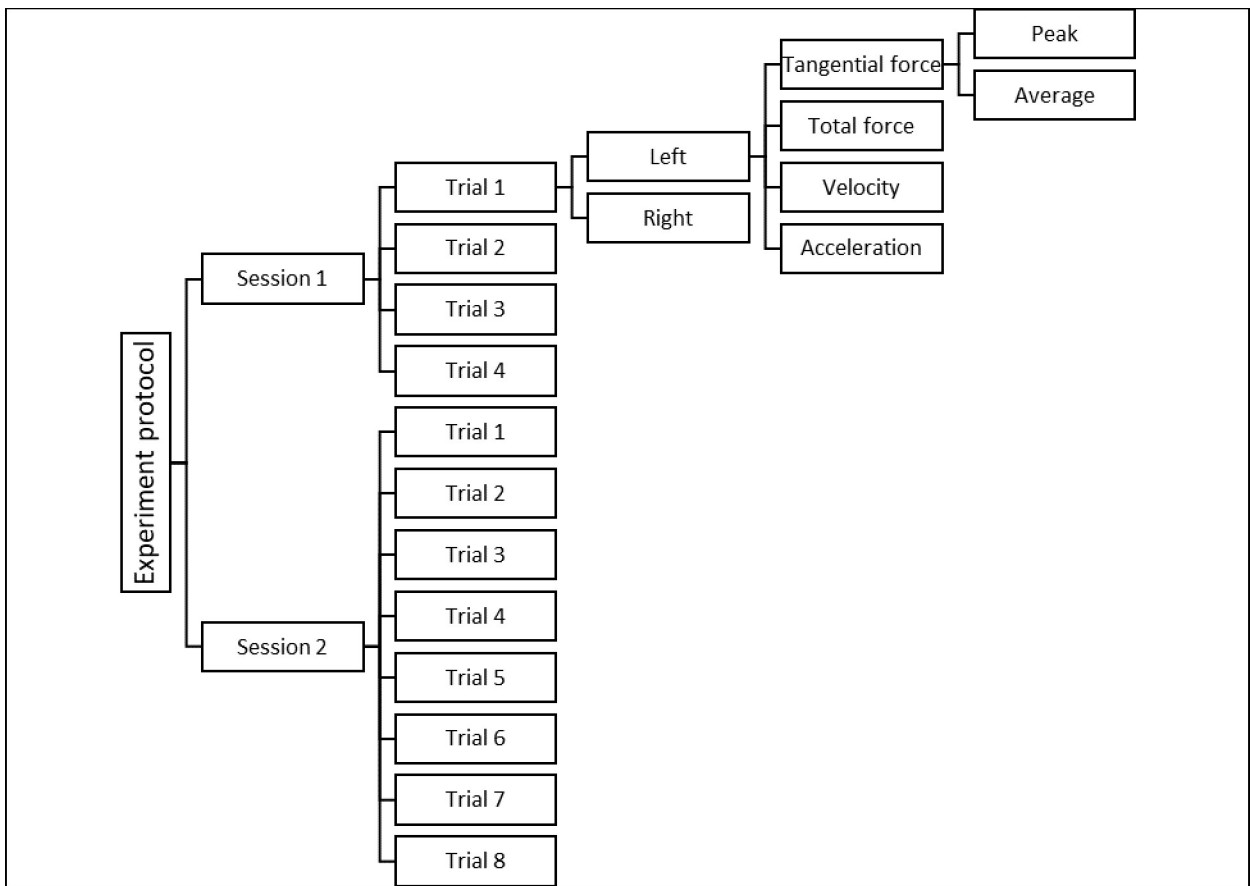

**Fig 1. Flow chart of the experiment including the measured variables.**

For this study, a standardized version of IAT was used [9], which is overall a 10 by 5 meters court; the cones on the horizontal lines are placed 2.5 meters in between, and on the vertical line they are positioned 3.3 meters apart (Fig 2). Since completion time is not a great concern for wheelchair users when accomplishing wheelchair manoeuvres, participants were instructed to propel their wheelchair through the IAT path at their comfortable speed. They were asked to read their heart rate just before and just after finishing the test, using a Fitbit Charge HR (wireless heart rate and activity wristband). The wheelchair that was used for the experiments was Quickle GP and was equipped with two 24-inch SMART$^{Wheel}$s, the validity of which for measuring propulsion forces has been shown by Asato et al. [19]. The subjects were video-captured while performing the test.

## Data analysis

Statistical analyses were performed using SPSS (IBM® SPSS® Statistics Premium GradPack 24 for Windows) and the alpha level (critical significance) was set to 0.05.

At each trial, velocity and force information were recorded using SMART$^{Wheel}$. Then, using a custom code the authors wrote in MATLAB, two categories of parameters were derived for each trial: data series and single-value parameters.

**Data-series parameters.** Sixteen variables were derived at thirty-two measurement points (Fig 3) throughout the IAT path: peak and average magnitude of tangential and total force,

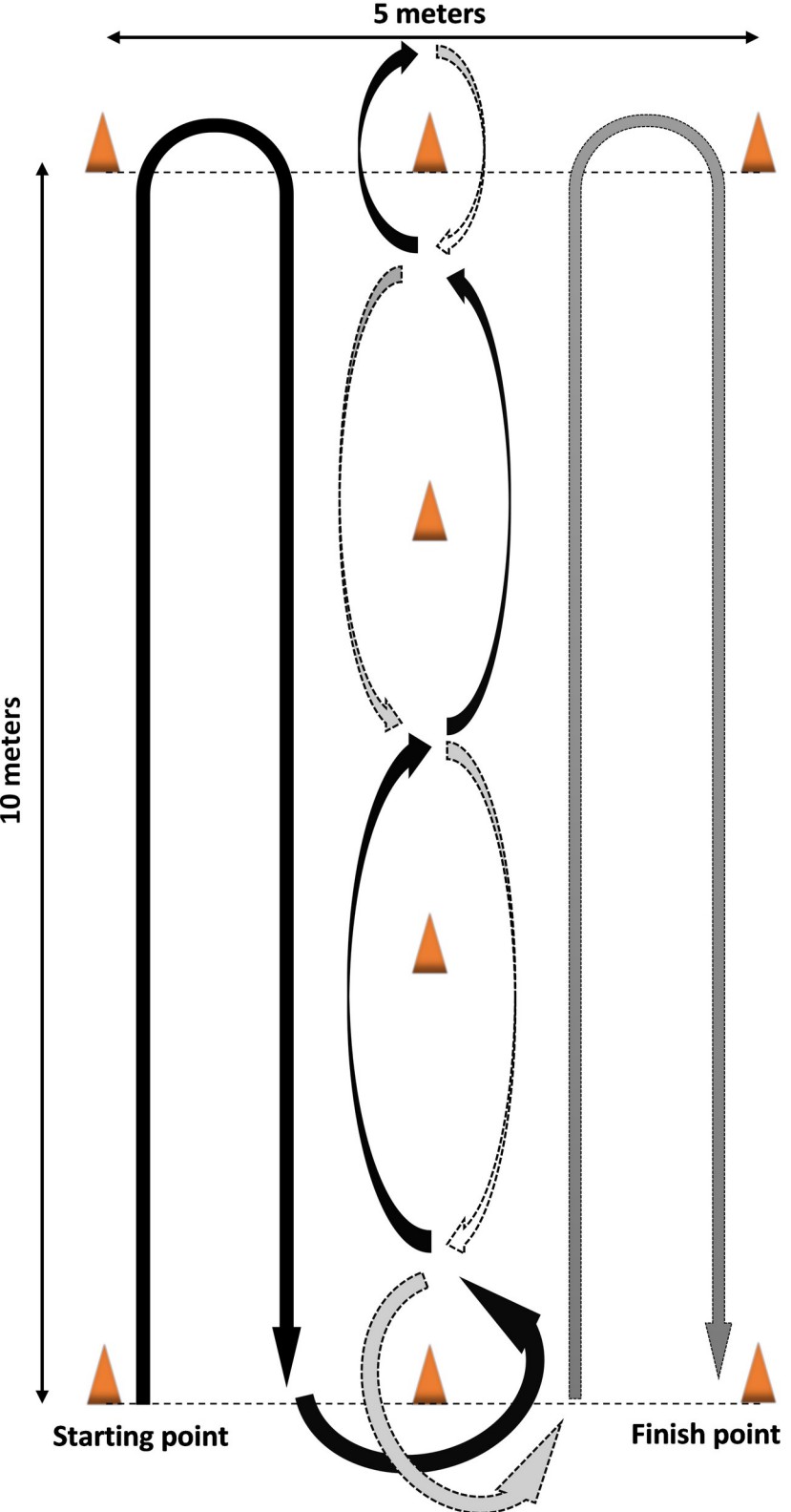

**5 meters**

**10 meters**

**Starting point**

**Finish point**

**Fig 2. Illinois Agility Test [9].**

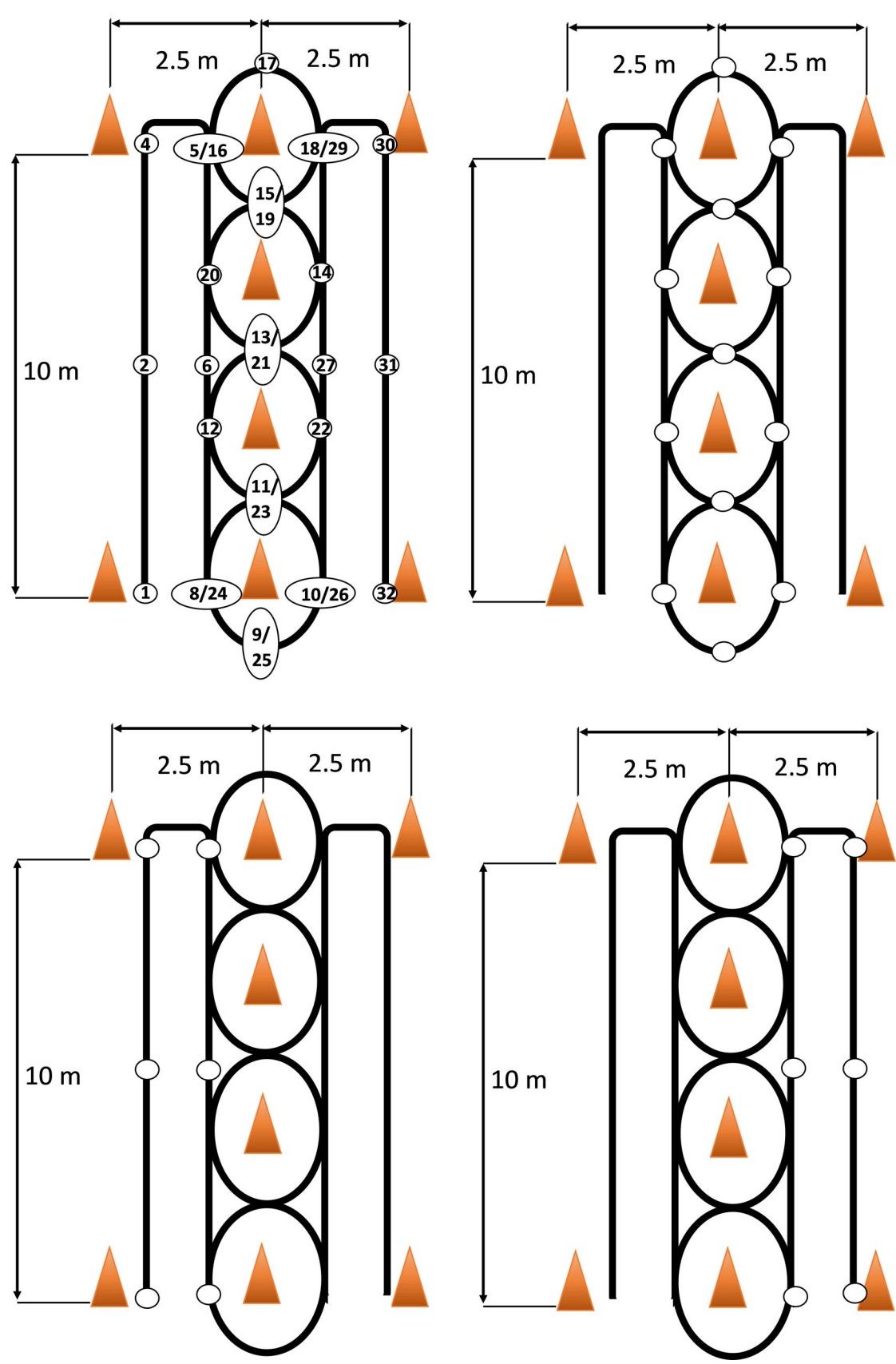

**Fig 3.** The measurement points considered throughout the IAT path; (a) shows the full path, (b) shows the manoeuvring part of the path, and (c) and (d) specify the straight-line propulsion part.

velocity, and acceleration, both for the left and the right side (Fig 1). Twenty-nine measurement points are shown in Fig 3A. Three other points (points number 3,7, and 28) pertained to the last push in the Straight-Line Propulsion (SLP) parts of IAT, just before turning pushes start (Fig 3C and 3D). Since participants could start turning anywhere between the two neighboring measurement points, e.g. 2 and 4, depiction of these points (3, 7, and 28) was not feasible. The guidelines used to find the measurement points are presented in Table 1.

To find the measurement points, we could either use Optitrack (motion analysis) or regular cameras to visually check the video recordings. Optitrack (NaturalPoint, Inc., USA, ARENA Motion Capture- Software: Motive_Tracker- Hardware: Prime 17W and S250e cameras) is a very high precision motion capture system and provides very accurate data, but due to prolonged nature of the tests and the number of repetitions of the tests, using Optitrack required an unrealistic effort to track each test frame by frame and to run custom algorithms to find the measurement points. Whereas, using video recordings it was possible to reduce the analysis time by a factor of 5. Even better, it prevented possible misinterpretations of participant's movement., e.g. when the participant makes a substantial turn but is still on SLP and has not yet started the real turn.

To confirm that the video recording method was sufficiently accurate, for one subject, the test was recorded using both Optitrack and regular cameras. Then, the measurement points were found using the trajectory obtained from the Optitrack data and the guidelines presented in Table 1. These points were also obtained visually by reviewing the video record of the test frame by frame (frame rate of 30 fps) using Avidemux 2.6.13. The points found using each method were then analyzed using SPSS to find the agreement ICC and Pearson Correlation Coefficient (PCC). The data obtained using the two techniques are shown in Fig 4. The PCC and ICC were found to be 100% between points obtained from Optitrack and camera (Confidence Interval (CI) = 99.8%-100%). Therefore, the video-tape method was selected in order to reduce analysis time.

After finding all the measurement points for all variables and all participants-trials, variables pertaining to each trial were averaged over all subjects. Then, the data for 32 points of

**Table 1. Method of finding the location of measurement points.**

| Point | Method of finding the coordinates |
|---|---|
| 1 | The first data set |
| 4 | At the time the yaw angle of body passes -7 deg* |
| 2 | When y equals to average of y coordinates of point 1 and 4 |
| 5 | At the time the yaw angle of body passes -173 deg |
| 8 | At the time the yaw angle of body returns to -173 deg and passes it |
| 6 | When y equals to average of y coordinates of point 5 and 8 |
| 9,11,13,15,17 | When x equals to 0** |
| 10,12,14,16 | At the extrema of y |

Only information for half of the points are provided here; the other half of the points use similar methods, due to symmetry. For point numbers please refer to Fig 3A.

* (-7) degrees is the inverse tangent of ((2.5m/2)/10m) and is the threshold after that we are sure turning has been started.

** The origin of the coordinate system is placed at the middle cone at the starting and finish line.

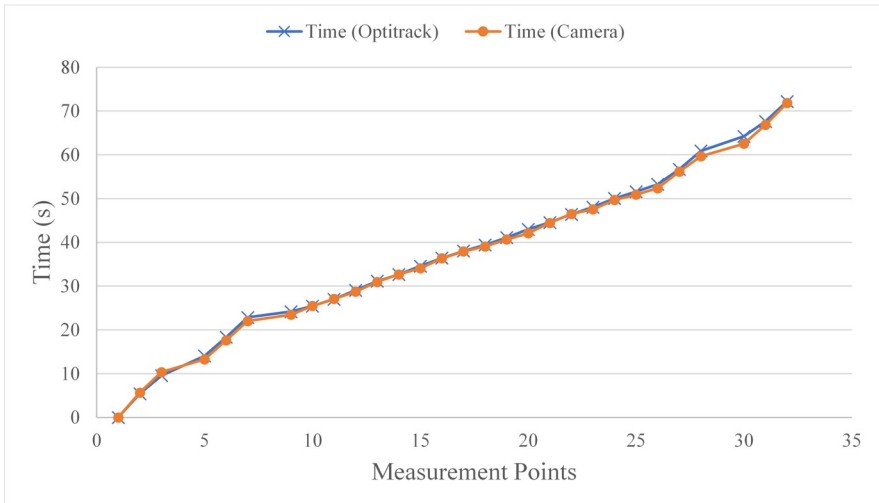

**Fig 4. The time one subject passed the measurement points obtained from Optitrack and camera records.**

measurement for each variable were averaged over all trials in each session. As a result, 32 averaged measurement points were achieved for each session-variable.

Data were checked for normality first, using Shapiro-Wilk test. Data analysis was performed using SPSS and Shapiro-Wilk test is recommended for testing normal distribution in SPSS [20]. Since parametric methods have higher research power than non-paramedic methods, where data did not follow normal distribution, we attempted normalization using data transformation. Considering the relatively small sample size here, we did not want to compromise the power of the study by using non-parametric methods. Therefore, in the event a variable had a non-normal distribution, the Two-Step [21] method was used to transform the data of that variable for both sessions. In the Two-Step method, we first transform the original values to fit uniform distribution and then analyzed the normalized data.

The test-retest reliability of the 16 variables was found using absolute agreement Intra-Class Correlation coefficient (ICC model (2,4)) between the scores of the first and second sessions. Also, the Standard Error of Measurements (SEM) was determined for each variable, using this formula [22]:

$$SEM = SD \times \sqrt{(1 - ICC)} \tag{3}$$

Where SD is the pooled standard deviation of all data for each variable and ICC is the intra-class correlation coefficient obtained for that variable.

**Single-value parameters.** Thirty-seven variables were derived as single-values for each trial-subject, which are significant parameters in the biomechanics of wheelchair propulsion, and decisive indices of similarity of the tasks repeated over different sessions. These variables are classified into four groups:

Group I: time used to finish:

- The whole trial (t_Tot (s))

- The straight-line part (t_SLP (s))

- The manoeuvring part (t_Mnv (s))

Group II: Participant's average cadence (frequency of pushing) when performing one IAT trial, for the right and the left wheel (Cadence (1/s).

Group III: Push parameters for the right and the left side:

- Push starting angle (Starting Angle (Deg))

- Push length (Push Length (Deg))

- Number of pushes used to perform the first U-turn in IAT (U-TURNS1)

- Number of pushes used to perform the second U-turn in IAT (U-TURNS2)

- Number of pushes used to perform the SLP part of IAT (ALL-SLP)

- Number of pushes used to go around the cones on the left (GOING-AROUND-CONES-ON-LEFT)

- Number of pushes used to go around the cones on the right (GOING-AROUND-CONES-ON-RIGHT)

Group IV: Sum of forces applied by the wheelchair user on the right and the left side:

- Sum of tangential forces applied on the rim during the whole path (SigmaFt_(TOT) (N))

- Sum of tangential forces applied on the rim during the SLP part (SigmaFt_(SLP) (N))

- Sum of tangential forces applied on the rim during the manoeuvring part (SigmaFt_(MNV) (N))

- Sum of absolute tangential forces applied on the rim during the whole path (SigmaFt_ABS (TOT) (N))

- Sum of absolute tangential forces applied on the rim during the SLP part (SigmaFt_ABS (SLP) (N))

- Sum of absolute tangential forces applied on the rim during the manoeuvring part (SigmaFt_ABS(MNV) (N))

- Sum of total forces applied on the rim during the whole path (SigmaFtot_(TOT) (N))

- Sum of total forces applied on the rim during the SLP part (SigmaFtot_(SLP) (N))

- Sum of total forces applied on the rim during the manoeuvring part (SigmaFtot_(MNV) (N))

Data analysis for the single-value parameters was also performed using SPSS. Firstly, for each subject, all variables were averaged over all trials in each session. Secondly, these averaged data were checked for normality. Thirdly, variables that did not have normal distribution were transformed using the Two-Step method [21]. Fourthly, one one-way MANOVA (Multivariate analysis of variance) was performed for variables belonging to each group (four MANOVAs in total). To check the assumption of homogeneity of variances (a necessary condition for performing MANOVA [23]), when computing Box's Test of equality of covariance matrices was not possible because of non-singular cell covariance matrices in the data (being caused by occasional missing data), the inter-item covariance matrix of all variables belonging to that group was developed and variances were visually checked for having close values.

## Results

### Data series parameters

For data series variables, 10 out of 32 (16 variables for each session) needed a transformation to comply with normality. Since some of these non-normal variables belonged to only one session

**Table 2. ICC, lower and upper band of confidence interval, as well as standard error of measurement (SEM) for the sixteen variables of data series parameters: Average and peak values of velocity, acceleration, tangential and total force, each for the left and right.**

| | Mean | SD | ICC | CI_LB | CI_UB | SEM |
|---|---|---|---|---|---|---|
| Avg Vel_L (m/s) | 0.77 | 0.025 | 0.88 | -0.07 | 0.97 | 0.01 m/s |
| Peak Vel_L (m/s) | 0.86 | 0.02 | 0.83 | -0.07 | 0.96 | 0.01 m/s |
| Avg Ft_L (N) | 6.16 | 32.09 | 0.96 | 0.82 | 0.99 | 6.42 (N) |
| Peak Ft_L (N) | 29.09 | 146.14 | 0.96 | 0.71 | 0.99 | 32.68 (N) |
| Avg Ftot_L (N) | 16.07 | 8.28 | 0.89 | 0.69 | 0.96 | 2.75 (N) |
| Peak Ftot_L (N) | 39.82 | 56.83 | 0.94 | 0.54 | 0.98 | 15.04 (N) |
| Avg Acc_L (m/s$^2$) | -0.006 | 0.009 | 0.98 | 0.97 | 0.99 | 0.001 m/s$^2$ |
| Peak Acc_L (m/s$^2$) | 0.295 | 0.016 | 0.93 | 0.04 | 0.98 | 0.006 m/s$^2$ |
| Avg Vel_R (m/s) | 0.724 | 0.026 | 0.96 | 0.19 | 0.99 | 0.01 m/s |
| Peak Vel_R (m/s) | 0.82 | 0.019 | 0.91 | -0.06 | 0.98 | 0.01 m/s |
| Avg Ft_R (N) | 2.43 | 35.18 | 0.98 | 0.95 | 0.99 | 6.09 (N) |
| Peak Ft_R (N) | 22.45 | 121.32 | 0.95 | 0.77 | 0.98 | 29.72 (N) |
| Avg Ftot_R (N) | 14.93 | 5.81 | 0.42 | -0.22 | 0.76 | 4.43 (N) |
| Peak Ftot_R (N) | 37.02 | 71.01 | 0.73 | -0.15 | 0.93 | 37.58 (N) |
| Avg Acc_R (m/s$^2$) | -0.019 | 0.013 | 0.98 | 0.97 | 0.99 | 0.002 m/s$^2$ |
| Peak Acc_R (m/s$^2$) | 0.26 | 0.019 | 0.97 | 0.93 | 0.98 | 0.004 m/s$^2$ |

and some of them were repeated in both sessions, 14 variables were transformed in total. To check the agreement of data, the comparison was made between the two data series which were based on the same construct: either the raw value or the transformed function of it.

Table 2 details the results achieved for the data series parameters. The average ICC of these variables was 89%. Also, the average 95% confidence interval was (44%-96%) (See Table 2 for details).

## Single-value parameters

The results of single-value parameters of eleven subjects are presented in this part. After checking for compliance with normal distribution, 15 out of 37 variables (38%) were shown to need variable transformation, which was done using the Two-Step method [21]. The results of each group of single-value parameters are presented under the respective headings.

**Group I: Time.** The statistical details of results for the time taken for subjects to finish the whole IAT path and the SLP and manoeuvring part of it are presented in Table 3.

The significance of Box's test of equality of covariance matrices for these variables was obtained as 0.307 and this does not provide enough evidence to reject the null hypothesis of Box's test which says the covariance of matrices are equal, so this MANOVA assumption is also met. Df2 for this group of parameters was 19.

The significance of MANOVA results was 0.441 (Pillai's Trace method) which cannot reject the null hypothesis that says the time taken for the subjects to finish different parts of IAT is statistically the same at the two sessions (similarity was not rejected- favorable for our study).

**Table 3. Detailed statistical results for the time taken to finish different tasks.**

| | Mean (SD) (Sec) | | F | Effect size ($\eta^2$) | P-Value |
|---|---|---|---|---|---|
| | Session 1 | Session 2 | | | |
| **Whole path** | 100.37 (12.41) | 90.14 (21.57) | 1.584 | 0.081 | 0.224 |
| **Manoeuvring part** | 46.54 (7.07) | 41.57 (9.72) | 1.636 | 0.083 | 0.217 |
| **SLP part** | 46.58 (16.66) | 39.75 (10.12) | 2.287 | 0.113 | 0.148 |

**Group II: Cadence.** The mean and standard deviation of the left and right cadence is presented in Table 4. Box's test of equality of covariance matrices was not significant (sig = 0.564) which means the assumption of homogeneity of variances and covariances is met. Df2 for this group of parameters was 18. Pillai's Trace significance was 0.428 which again shows the MANOVA results for cadence were not significant (P-value<0.05) and therefore the similarity of the push frequencies of the left and right side was not rejected statistically between session 1 and session 2.

**Group III: Push parameters.** To find the main results obtained for push parameters during IAT task performed by eleven able-bodied wheelchair users at two different days see Table 5. For this group of parameters, the Box's test of equality of covariance matrices returned no results due to non-singular cell covariance matrices in the data which are caused by missing data of one person in the first session. Thus, the inter-item covariance matrix of all variables belonging to that group was visually checked for having close values. It was observed that in 85% of all cases the data were fairly close and therefore it was concluded that the assumption of homogeneity of variances and covariances is met. Df2 for this group of parameters was 17.

Regarding the MANOVA results for this group, Pillai's Trace significance was obtained as 0.378 which means, here again, there is no statistically significant difference between push parameters in session one compared to session 2.

## Group IV: Sum of forces

Table 6 details the main outcomes obtained regarding the sum of forces applies on the right and left wheel when performing IAT by non-experienced wheelchair users.

Similar to parameters of group 3, the Box's test of equality of covariance matrices returned no results, and checking the inter-item covariance matrix of all variables showed that 93.2% of them had relatively close values. Therefore, it was concluded that the assumption of homogeneity of variances and covariances was met. Df2 for this group of parameters was 15.

Also, in the MANOVA results for this group, Pillai's Trace significance was 0.356 which means the two sessions were not statistically different in terms of the sum of forces applied on the wheels.

## Discussion

In this study, we recruited eleven able-bodied subjects to participate in two sessions of wheelchair manoeuvring with the aim of studying the reliability of a standard agility test adapted for wheelchair users. A comprehensive assessment was performed considering fifty-three parameters in total, out of which, sixteen were obtained as data series throughout the IAT. Considering ICC = 0.6 as the minimum acceptable ICC [18], all the data series variables studied were shown to be reliable when performing IAT by wheelchair users, except for the average total force of the right side. Hence, 94% of these variables were shown to have a good to excellent reliability. The average total force on the right wheel was the one with the lowest reliability (42%), while the same variable for the left wheel had very good repeatability (89%) and other force variables also had good to excellent repeatability. Therefore, we suspect that there may have been fundamental differences in the force application methods when participants

**Table 4. Detailed statistical results for push frequency (cadence) for the left and right wheel.**

| | Mean (SD) (1/Sec) | | F | Effect size ($\eta^2$) | P-Value |
|---|---|---|---|---|---|
| | Session 1 | Session 2 | | | |
| **Left Cadence** | 0.86 (0.040) | 0.87 (0.041) | 0.269 | 0.014 | 0.610 |
| **Right Cadence** | 0.85 (0.035) | 0.84 (0.038) | 0.083 | 0.004 | 0.776 |

**Table 5. Detailed statistical results for push parameters of the left and right wheel.**

| | Mean (SD) | | F | P-Value | Effect size ($\eta^2$) |
|---|---|---|---|---|---|
| | Session 1 | Session 2 | | | |
| Left Starting Angle* (Deg) | -9.72 (0.23) | -9.29 (0.8) | 1.709 | 0.209 | 0.091 |
| Right Starting Angle* (Deg) | -9.6 (0.26) | -9.5 (0.39) | 0.424 | 0.524 | 0.024 |
| Left Push Length (Deg) | 18.63 (3.74) | 21.12 (4.66) | 2.394 | 0.140 | 0.123 |
| Right Push Length (Deg) | 17.59 (3.96) | 20.87 (5.11) | 3.028 | 0.100 | 0.151 |
| Left U-turn 1 | 4.9 (0.76) | 4.2 (0.91) | 4.930 | 0.040 | 0.225 |
| Right U-turn 1 | 4.2 (1.3) | 3.7 (0.96) | 1.742 | 0.204 | 0.093 |
| Left U-turn 2 | 5.1 (1.53) | 4.6 (1.06) | 0.740 | 0.402 | 0.042 |
| Right U-turn 2 | 4.9 (1.84) | 3.8 (1.54) | 4.817 | 0.042 | 0.221 |
| Left ALL-SLP | 37 (9.3) | 31.5 (8.85) | 2.826 | 0.111 | 0.143 |
| Right ALL-SLP | 36 (8.5) | 31.6 (8.89) | 2.236 | 0.153 | 0.116 |
| Left wheel-going around cones on left side | 4.6 (1.08) | 3.8 (0.92) | 3.965 | 0.063 | 0.189 |
| Right wheel-going around cones on left side | 4.4 (0.97) | 3.9 (0.96) | 1.307 | 0.269 | 0.071 |
| Left wheel-going around cones on right side | 4.7 (1.12) | 4.2 (1.08) | 1.242 | 0.281 | 0.068 |
| Right wheel-going around cones on right side | 4.5 (0.86) | 3.9 (0.98) | 2.470 | 0.134 | 0.127 |

*-0 degree corresponds to the person's longitudinal axis and negative angles represent the participants extending their hands backward.

repeated the test over two visits. Since the devices were calibrated before starting the experiments, we do not believe that there was an instrumentation issue. Furthermore, performing MANOVA on each group of single-value parameters showed that there were no statistically meaningful differences between all of these thirty-seven variables between sessions one and two, which, in turn, backs the reliability of IAT for wheelchair users.

**Table 6. Detailed statistical results for push parameters of the left and right wheel.**

| | Mean (SD) (KN) | | F | P-Value | Effect size ($\eta^2$) |
|---|---|---|---|---|---|
| | Session 1 | Session 2 | | | |
| Left SigmaFt_Tot | 170.84 (101.54) | 111.39 (39.21) | 0.265 | 0.614 | 0.017 |
| Right SigmaFt_Tot | 55.86 (14.05) | 51.77 (17.28) | 0.134 | 0.720 | 0.009 |
| Left SigmaFt_Mnv | 67.51 (46.59) | 39.12 (21.45) | 0.492 | 0.494 | 0.032 |
| Right SigmaFt_Mnv | 36.07 (5.87) | 34.13 (7.53) | 0.251 | 0.623 | 0.016 |
| Left SigmaFt_SLP | 77.23 (42.3) | 55.95 (15.3) | 0.081 | 0.780 | 0.005 |
| Right SigmaFt_SLP | 30.71 (5.2) | 30.73 (11.65) | 0.000 | 0.983 | 0.000 |
| Left SigmaFt_ABS(Tot) | 252.92 (69.18) | 224.67 (28.16) | 0.521 | 0.482 | 0.034 |
| Right SigmaFt_ABS(Tot) | 192.56 (26.4) | 196.86 (33.39) | 0.030 | 0.866 | 0.002 |
| Left SigmaFt_ABS(Mnv) | 124.76 (27.41) | 115.6 (17.03) | 0.183 | 0.675 | 0.012 |
| Right SigmaFt_ABS(Mnv) | 93.26 (15.61) | 91.5 (14.48) | 0.074 | 0.789 | 0.005 |
| Left SigmaFt_ABS(SLP) | 93.18 (35.15) | 79.51 (14.54) | 0.079 | 0.782 | 0.005 |
| Right SigmaFt_ABS(SLP) | 62.82 (6.81) | 66.15 (10.54) | 0.217 | 0.648 | 0.014 |
| Left SigmaFtot_Tot | 419.33 (200.95) | 323.1 (199.26) | 0.101 | 0.756 | 0.007 |
| Right SigmaFtot_Tot | 350.81 (205.76) | 393.4 (108.01) | 0.027 | 0.872 | 0.002 |
| Left SigmaFtot_Mnv | 206.21 (86.46) | 164.16 (92.51) | 0.000 | 0.999 | 0.000 |
| Right SigmaFtot_Mnv | 164.79 (93.91) | 181.55 (49.65) | 0.316 | 0.582 | 0.021 |
| Left SigmaFtot_SLP | 161.26 (96.53) | 118.77 (80.89) | 0.178 | 0.679 | 0.012 |
| Right SigmaFtot_SLP | 141.18 (83.56) | 155.15 (50.64) | 0.000 | 0.998 | 0.000 |

It is worth mentioning that although the number of variables tested here is large, there is no need to use the Bonferroni correction, as this is used when we are worried that the "changes (differences) detected" are not real and could have occurred by chance. Whereas in this study, we were interested to show "no differences" among the two sessions. In fact, *not considering* the Bonferroni correction adds to the power of study here, as the null hypotheses are *maintained* by Alpha level of 0.05. Usually, the Bonferroni correction is used, to adjust the alpha level, so that a P-value (0.04, for example), does not result in claiming a significant change. Whereas in our case, the case is the opposite, and rejection by a lower alpha level is favorable, not restricting. Thus, using Bonferroni Correction is pointless, and even worse, it decreases the study power.

In this study, we recruited able-bodied subjects instead of experienced wheelchair users. This helped with the recruitment, but more importantly, supported the main purpose of this study which was studying the reliability of IAT rather than the wheelchair users themselves. Experienced wheelchair users are much more consistent in the way they individually propel their wheelchair compared to non-wheelchair-users. We believed that if we could show that non-experienced wheelchair users were very consistent in performing a wheelchair manoeuvring task when repeating it 10 days later (on average), this would provide stronger evidence that this test (IAT) can be used for both experienced and non-wheelchair users, as a reliable assessment tool.

Experienced wheelchair users are more efficient than non-wheelchair-users [24]. They tend to use longer pushes with lower peaks both to prevent overuse injuries and to increase efficiency [25]. Would this potentially increase inconsistency in the propulsion technic? It is unlikely, but further research is warranted recruiting real wheelchair users to be conclusive about the reliability of IAT for real wheelchair users.

Differences in wheelchair manoeuvring that may be associated with clinical factors (level of spinal injury, upper extremity injury) could be thought of as a possible source of propulsion variance the biomechanical parameters of interest during manoevering. However, these differences between participants do not violate our assumption as we believe within-subject consistency is higher among this population. Although the between-subject variance could be higher in wheelchair users, it does not harm our assumption that recruiting non-wheelchair-users provides stronger evidence for the reliability of IAT for wheelchair users.

The single-value parameters measured and analyzed for this study were selected based on clinical and practical relevance and importance. These measurements are frequently needed by both clinicians and engineers who work with or for wheelchair users. For instance, cadence and push length are important parameters that clinicians work on with newly injured wheelchair users [26], as they work effectively toward propulsion efficiency [27] which, in turn, plays a very important role in secondary injuries occurred to wheelchair users, as propulsions with higher efficiency would incorporate less total force to accomplish a given task [25], and that works against overuse of shoulder. Also, tangential and total forces applied on the rim are both effective on propulsion efficiency [25, 28] and determinative when measuring activities of wheelchair users [29]. Moreover, the starting angle is an important factor when adjusting the wheelchair axle for the stability of the wheelchair, and preventing secondary injuries [30]. The factors reported in this paper are therefore both clinically and technically informative and we observed that the comfortable speed when manoeuvring with a wheelchair was 0.85 m/s which is considerably lower than 1.1 m/s [1, 31] or 1.4 to 1.5 m/s (equal to 83.4 to 90.7 m/min) [32] which is the comfortable speed that has been reported before for performing straight-line wheelchair propulsion. Table 4 is also particularly useful as it displays valuable push parameters when performing IAT. One interesting observation in this table, for instance, is that when performing a rather complex manoeuvring task, participants did not show any indication of having more pushes on their dominant hand. However, although statistically the number of pushes in completing different parts of IAT was not different in session one compared to

session two, we can see a consistent decrease in the number of pushes in session two relative to session one which could be clinically meaningful and a sign that participants used fewer pushes as they began to master the task.

Furthermore, Table 2 shows the average velocity, acceleration, and forces participants used in one average push. However, we have also reported the actual and absolute sum of forces applied by the participants for finishing different parts of IAT. Note that this is the summation of forces at all frames in each part that is being studied (240 Hz). The clinical importance of these parameters is that eventually, we are interested in the amount of force a wheelchair user needs to apply when performing different tasks. These data can be used as a reference for future works where having an estimate for forces applied during wheelchair propulsion is necessary, such as developing assistive devices, activity monitors, or injury prevention apparatus for wheelchair users. They can also be used where clinicians need to know which activities require more force and hence increase the risk of overuse injuries.

## Conclusion

We were able to assess the reliability of a standard agility test, IAT, so that now it can be used in research studies focusing on wheelchair manoeuvring and also in training assessments of wheelchair-sport athletes.

We showed that 94% of the important biomechanical parameters are statistically reliable when wheelchair users perform IAT, and we furthermore showed that 37 other biomechanical variables do not statistically differ when the test is repeated after a reasonably lengthy period of time.

Other clinical or practical implications were also provided, such as the average speed of wheelchair manoeuvring which was 0.85, about 57% to 77% of the average speed of SLP wheelchair propulsion reported elsewhere [1, 31, 32].

Future work on wheelchair manoeuvre can use findings of this study on the reliability of IAT. Also, the single-value parameters, in particular, can be helpful to clinicians and engineers in developing injury-prevention strategies and devices.

## Acknowledgments

The authors would like to thank the participants of this study who kindly agreed to devote their time for conducting these experiments.

## Author Contributions

**Conceptualization:** Zohreh Salimi, Martin William Ferguson-Pell.

**Data curation:** Zohreh Salimi, Martin William Ferguson-Pell.

**Formal analysis:** Zohreh Salimi.

**Investigation:** Zohreh Salimi.

**Methodology:** Zohreh Salimi, Martin William Ferguson-Pell.

**Project administration:** Zohreh Salimi.

**Resources:** Zohreh Salimi, Martin William Ferguson-Pell.

**Software:** Zohreh Salimi.

**Supervision:** Martin William Ferguson-Pell.

**Validation:** Zohreh Salimi.

**Visualization:** Zohreh Salimi, Martin William Ferguson-Pell.

**Writing – original draft:** Zohreh Salimi.

**Writing – review & editing:** Martin William Ferguson-Pell.

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
