## [Decision Letter · Decision Letter 0]

14 May 2020

PONE-D-20-08803

Investigating the test-retest reliability of Illinois Agility Test for wheelchair users

PLOS ONE

Dear Dr. Salimi,

Thank you for submitting your manuscript to PLOS ONE. After careful consideration, we feel that it has merit but does not fully meet PLOS ONE’s publication criteria as it currently stands. Therefore, we invite you to submit a revised version of the manuscript that addresses the points raised during the review process.

We would appreciate receiving your revised manuscript by Jun 28 2020 11:59PM. To enhance the reproducibility of your results, we recommend that if applicable you deposit your laboratory protocols in protocols.io, where a protocol can be assigned its own identifier (DOI) such that it can be cited independently in the future. For instructions see: http://journals.plos.org/plosone/s/submission-guidelines#loc-laboratory-protocols

We look forward to receiving your revised manuscript.

Kind regards,

Yih-Kuen Jan, PhD

Academic Editor

PLOS ONE

Journal Requirements:

2. In your Methods section, please state where the participants were recruited for your study.

'No- The funders had no role in study design, data collection and analysis, decision to publish, or preparation of the manuscript.'

4. We noted in your submission details that a portion of your manuscript may have been presented or published elsewhere:

'This paper is prepared from the results of my PhD thesis and is included as Chapter 7 in my dissertation (Salimi, 2017).'

Please clarify whether this publication was peer-reviewed and formally published. If this work was previously peer-reviewed and published, in the cover letter please provide the reason that this work does not constitute dual publication and should be included in the current manuscript.

Reviewers' comments:

Reviewer's Responses to Questions

**Comments to the Author**

1. Is the manuscript technically sound, and do the data support the conclusions?

Reviewer #1: Partly

Reviewer #2: No

2. Has the statistical analysis been performed appropriately and rigorously? 

Reviewer #1: No

Reviewer #2: No

3. Have the authors made all data underlying the findings in their manuscript fully available?

Reviewer #1: Yes

Reviewer #2: Yes

4. Is the manuscript presented in an intelligible fashion and written in standard English?

Reviewer #1: No

Reviewer #2: Yes

5. Review Comments to the Author

Reviewer #1: General Comments:

•This paper investigated maneuverability in manual wheelchair users using a validated agility (IAT). This test has been validated in able-bodied athletes, and one study validated it time to finish in manual wheelchair athletes. This study should clarify its intent to validate biomechanical factors used within the test; not specifically the test, itself.

•Numerous grammatical, citation, and reading flow issues throughout. Several instances throughout the text where sections can be shortened and more “to the point.”

•Authors should reevaluate the statistical tests employed in this study and the interpretations that follow.

Introduction:

•Line 52: Sentences with only one supporting reference should acknowledge the reference at the END of the sentence rather than in the middle of the sentence.

•Lines 50-52: Reference is needed for these statements.

•Lines 55-57: Consider revising to read “…excessive forces exerted on the structures around the shoulder during wheelchair propulsion…” Reference is needed for this statement.

•Line 71, “We could not find any publications that have shown the reliability of an agility test for wheelchair users” is contradicted by Lines 79-84. Williams et al. did investigate the validity of IAT within the context of time to finish, and deemed it validated. While it may be true that they did not investigate the validity of “other important biomechanical factors,” it is important to remember why the IAT was created and how it is used: agility (presumably determined by time to finish as mentioned in Lines 88-90). It appears that this paper aims to validate “other important biomechanical factors” using the IAT; this should be clearer in the Title, Abstract, and Introduction as the main outcome of the IAT (i.e. agility) has already been validated.

•Line 93: Specify “the latter characteristic”

•Lines 94-97: Consider revising the purpose statement based on the fourth comment above.

Methods:

•Lines 125-127: Is the Fitbit Charge HR a validated tool for measuring heart rate? If so, this tool may be useful in the able-bodied manual wheelchair users in this study; however, heart rate is not a validated metric in other manual wheelchair users, such as individuals with spinal cord injury. Translation of this methodology to manual wheelchair users with spinal cord injury/disease may not be appropriate.

•Lines 135-138: Error message should not likely be included in text (“Error! Reference source not found”)

•Line 139: Numbers 3,7, and 28 are not pictured in Figure 3. Is there a reason for this?

•Lines 148-172: Explanation and justification for using standard camera video over Optitrack should be much shorter. Additionally, this decision was based off one subject’s data. Why were additional subjects’ data not included to make this decision more robust?

•Lines 177-179, 226-228: How were data assessed for normality? If data did not fit a normal distribution, why were data transformed to fit a normal distribution instead of using non-parametric statistical tests? Briefly describe the Two-Step method for readers who are not familiar with this test.

•The same study subjects performed the IAT at two timepoints roughly one week apart. Why was a MANOVA test conducted instead of a repeated measures MANOVA?

•No mention of significance criteria; alpha/p-values, effect sizes, etc.

Results:

•Line 238: Sentence should not start with a number.

•Line 242: Consider using “raw value” in place of “real variable.”

•The Results section should be written in a more technical manner using p-values, F-statistics, degrees of freedom, etc.

•The Results section should focus on the statistical outputs and interpretation of these outputs should be reserved for the Discussion section.

Discussion:

•Line 308: Sentence should not start with a number.

•Line 317: Careful using definitive terminology like “confirms” when interpreting results.

•Lines 318-328: Authors should reexamine their interpretation of a Bonferroni correction. Lines 318-319 suggest it IS NOT advantageous for this study to use a Bonferroni correction; however, in Lines 325-326, the authors claim IT IS advantageous for this study to use a Bonferroni correction. A Bonferroni correction adjusts the p-value used to determine significance based on the number of dependent variables.

•The methods of this paper are designed to control for Type 1 Error (rejecting a true null hypothesis); however, it does not control for Type II Error (accepting a false null hypothesis). This should be addressed.

•Lines 331-337: Research has found that pediatric and adult manual wheelchair users with pediatric-onset spinal cord injury employ highly variable upper extremity movement. While the exact mechanism for this is unknown, some attribute it to adaptation to prevent overuse injury during wheelchair propulsion. The authors’ interpretation that experienced manual wheelchair users use “more consistent” mechanics based on experience may not always be true.

•Line 349: Many of these metrics are determined as “clinically meaningful.” How so? Provide rationale or references that mention the metrics used in this study as “clinically meaningful.”

•Do the authors have any directions for future work based on these results?

Conclusion:

•This section should “wrap-up” the manuscript and highlight the results of this study. Consider adding a few more sentences to bring the manuscript to a succinct ending.

Reviewer #2: It is inappropriate to use Null Hypothesis Significance Testing to argue for equivalence or equality. As it is typically used (i.e., when the researcher hopes to reject the null hypothesis), NHST is designed to make it difficult for the researcher to get what they want. When the researcher is hoping to 'prove' the null, however, as is the case here, achieving the desired result using NHST is quite easy: just design a poor and/or underpowered study. Failing to reject the null hypothesis means either: 1) that the null is true (in the case of this study, the repeated measurements are equal), or 2) the null is false (the repeated measurements are not equal) but the experiment doesn't have enough power to reject it. In order to arrive at the first conclusion (equality) the authors must provide evidence against the second (an underpowered study). Unfortunately, the authors provide no evidence of high statistical power. In fact, using only 11 subjects in the study makes it likely that the study is underpowered. In order for the results of the study to be meaningful, I strongly suggest that the authors forgo the use of NHST-based procedures such as MANOVA to determine equivalence in favor of more appropriate methods such as Bayesian methods, confidence intervals (Loftus, 1996; Aberson, 2002), or noninferiority testing (Streiner, 2003). If the authors insist on using NHST to claim evidence of equality, I view it as critical to demonstrate that the study as ran had high power. In addition, effect size measures, F statistics, and their associated degrees of freedom should be included for all MANOVAs.

Aberson, C. (2002). Interpreting null results: Improving presentation and conclusions with confidence intervals. https://www.jasnh.com/a6.htm

Loftus, G. L. (1996). Psychology will be a much better science when we change the way we analyze data. Current Directions in Psychological Science, 5, 161-171.

Streiner, D. L. (2003). Unicorns do exist: A tutorial on "proving" the null hypothesis. Canadian Journal of Psychiatry, 48, 756-761.

6. PLOS authors have the option to publish the peer review history of their article (what does this mean?). If published, this will include your full peer review and any attached files.

Reviewer #1: Yes: Brooke A. Slavens, PhD

Reviewer #2: No

---

## [Author Response · Author response to Decision Letter 0]

28 Jun 2020

Editor: 

1 1. PLOS ONE's style requirements, including those for file naming Will do.

2 2. In your Methods section, please state where the participants were recruited for your study. Added.

3 3-a. Please clarify the sources of funding (financial or material support) for your study. List the grants or organizations that supported your study, including funding received from your institution. Canadian Foundation for Innovation (CFI 30852 Leading Edge Fund 2012) infrastructure grant

4 3-b. State what role the funders took in the study. If the funders had no role in your study, please state: “The funders had no role in study design, data collection and analysis, decision to publish, or preparation of the manuscript.” The funders had no role in study design, data collection and analysis, decision to publish, or preparation of the manuscript.

5 3.c. If any authors received a salary from any of your funders, please state which authors and which funders. No salary provided to any authors from this grant

6 3-d. If you did not receive any funding for this study, please state: “The authors received no specific funding for this work.” Please include your amended statements within your cover letter. The authors received no specific funding for this work. The CFI funding provided the capital equipment for the project.

7 4. We noted in your submission details that a portion of your manuscript may have been presented or published elsewhere. Please clarify whether this publication was peer-reviewed and formally published. If this work was previously peer-reviewed and published, in the cover letter please provide the reason that this work does not constitute dual publication and should be included in the current manuscript. My thesis chapter was not formally peer reviewed; it was examined for the purposes of meeting the requirements of a PhD. This paper is based on a chapter from my PhD thesis, and although it was examined, this did not constitute peer review and no specific recommendations were made at that time.

Reviewer #1: 

8 This paper investigated maneuverability in manual wheelchair users using a validated agility (IAT). This test has been validated in able-bodied athletes, and one study validated it time to finish in manual wheelchair athletes. This study should clarify its intent to validate biomechanical factors used within the test; not specifically the test, itself. With all due respect, this study aims in assessing the “reliability” of IAT, rather than validity. The paper you mentioned had checked the validity, which is out of scope of this study.

9 •Numerous grammatical, citation, and reading flow issues throughout. Several instances throughout the text where sections can be shortened and more “to the point.” Revised.

10 •Authors should reevaluate the statistical tests employed in this study and the interpretations that follow. Applied. Details are provided below, where questions point to details.

 Introduction: 

11 •Line 52: Sentences with only one supporting reference should acknowledge the reference at the END of the sentence rather than in the middle of the sentence. Displaced.

12 •Lines 50-52: Reference is needed for these statements. Added.

13 •Lines 55-57: Consider revising to read “…excessive forces exerted on the structures around the shoulder during wheelchair propulsion…” Reference is needed for this statement. Done.

14 •Line 71, “We could not find any publications that have shown the reliability of an agility test for wheelchair users” is contradicted by Lines 79-84. Williams et al. did investigate the validity of IAT within the context of time to finish, and deemed it validated. While it may be true that they did not investigate the validity of “other important biomechanical factors,” it is important to remember why the IAT was created and how it is used: agility (presumably determined by time to finish as mentioned in Lines 88-90). It appears that this paper aims to validate “other important biomechanical factors” using the IAT; this should be clearer in the Title, Abstract, and Introduction as the main outcome of the IAT (i.e. agility) has already been validated. Please note that Williams et al. demonstrated “validity” of IAT, whereas this paper aims assessing the “reliability” of it, for which we have stated that no publications were found.

15 •Line 93: Specify “the latter characteristic” Specified.

16 •Lines 94-97: Consider revising the purpose statement based on the fourth comment above. We respectfully clarified the misunderstanding in your fourth comment.

 Methods: 

17 •Lines 125-127: Is the Fitbit Charge HR a validated tool for measuring heart rate? If so, this tool may be useful in the able-bodied manual wheelchair users in this study; however, heart rate is not a validated metric in other manual wheelchair users, such as individuals with spinal cord injury. Translation of this methodology to manual wheelchair users with spinal cord injury/disease may not be appropriate. The heart rate measurement in this study was merely to ensure that the participants had enough rest time between the trials and was not included in the analyses. Since it was in addition to giving at least three minutes of rest time between the trials, it does not look essential to demonstrate the validity of Fitbit Charge HR. 

18 •Lines 135-138: Error message should not likely be included in text (“Error! Reference source not found”) This has happened when converting Word to Pdf. I will make sure it does not happen when submitting the revised version.

19 •Line 139: Numbers 3,7, and 28 are not pictured in Figure 3. Is there a reason for this? As it is explained in Line 140, points number 3,7, and 28 pertained to the last push in the straight-line propulsion parts of IAT, just before turning pushes were started. Since participants could start turning anywhere between the two neighboring measurement points, e.g. 2 and 4, depiction of these points (3, 7, and 28) was not feasible.

20 •Lines 148-172: Explanation and justification for using standard camera video over Optitrack should be much shorter. Additionally, this decision was based off one subject’s data. Why were additional subjects’ data not included to make this decision more robust? The explanation was shortened.

Regarding using data of one subject, as it was mentioned in the manuscript, analyzing Optitrack data for 120 frame/sec and an average of 120 sec/trial, if it is used for all 11 subjects, will take days, even months, to finish. The rational for comparing the video results to Optitrack results was to show the validity of video data, by testing a portion of data, which was confirmed by a very high correlation. 

21 •Lines 177-179, 226-228: How were data assessed for normality? If data did not fit a normal distribution, why were data transformed to fit a normal distribution instead of using non-parametric statistical tests? Briefly describe the Two-Step method for readers who are not familiar with this test. Normality was checked using Shapiro-Wilk test which is highly recommended for testing normal distribution in SPSS (Ghasemi & Zahediasl, 2012). Where data did not follow normal distribution, we tried our best to get them follow normal distribution through data transformation because parametric methods have higher research powers and considering the relatively small sample size here, we could not compromise power reduction. 

In the Two-Step method, we first transform the original values to fit uniform distribution and then to normal distribution. This is a roughly old statistical method that had kept outsider to many fields of science, but through the paper by Gary Templeton, 2011, it was re-introduced to scientific community to use it to their advantage.

These explanations are added to the main text for extra clarification.

22 •The same study subjects performed the IAT at two timepoints roughly one week apart. Why was a MANOVA test conducted instead of a repeated measures MANOVA? One of the assumptions of MANOVA is independency observations; Repeated Measures MANOVA should be used when there are repetitions of a measurement to adjust for dependency between measurements of the dependent variables. In this study, we wanted to show that dependent variables were similar from one session to the next one. So, we were better to first assume that they were different and thus independent, and then try to reject it. Therefore, we used MANOVA.

23 •No mention of significance criteria; alpha/p-values, effect sizes, etc. You are right. They are added to the manuscript now.

 Results: 

24 •Line 238: Sentence should not start with a number. Applied.

25 •Line 242: Consider using “raw value” in place of “real variable.” Applied.

26 •The Results section should be written in a more technical manner using p-values, F-statistics, degrees of freedom, etc. Added.

27 •The Results section should focus on the statistical outputs and interpretation of these outputs should be reserved for the Discussion section. Revised.

 Discussion: 

28 •Line 308: Sentence should not start with a number. Applied.

29 •Line 317: Careful using definitive terminology like “confirms” when interpreting results. Applied.

30 •Lines 318-328: Authors should reexamine their interpretation of a Bonferroni correction. Lines 318-319 suggest it IS NOT advantageous for this study to use a Bonferroni correction; however, in Lines 325-326, the authors claim IT IS advantageous for this study to use a Bonferroni correction. A Bonferroni correction adjusts the p-value used to determine significance based on the number of dependent variables. We are afraid you have not grasped the point raised in that paragraph. What we were trying to say was that usually the case is Bonferroni correction is used to adjust the alpha level, so that a P-value of 0.04 (for example) does not result in a claim for detecting a significant change. Whereas in our manuscript, the case is the opposite - a rejection by a lower alpha level is favorable, not restricting. Thus, using Bonferroni Correction is pointless and even worse, it decreases the study power.

We reworded the paragraph to be clearer.

31 •The methods of this paper are designed to control for Type 1 Error (rejecting a true null hypothesis); however, it does not control for Type II Error (accepting a false null hypothesis). This should be addressed. The method of sample size calculations for maintaining an acceptable research power, and thus acceptable chance of Type II error, is now added to the manuscript.

32 •Lines 331-337: Research has found that pediatric and adult manual wheelchair users with pediatric-onset spinal cord injury employ highly variable upper extremity movement. While the exact mechanism for this is unknown, some attribute it to adaptation to prevent overuse injury during wheelchair propulsion. The authors’ interpretation that experienced manual wheelchair users use “more consistent” mechanics based on experience may not always be true. Right. But as it was explained in Lines 336-345, note that this consistency comparison is between real experienced wheelchair users and non-wheelchair users. While within-subject variability for real wheelchair users could be relatively high, we need to ask how is the between-subject variability when they are compared to non-wheelchair-users. It seems self-evident that people who have no experience using a very different method of ambulation will be less consistent in wheelchair propulsion, overall.

33 •Line 349: Many of these metrics are determined as “clinically meaningful.” How so? Provide rationale or references that mention the metrics used in this study as “clinically meaningful.” The single-value parameters measured and analyzed for this study were selected based on clinical and practical relevance and importance. These are metrics that are frequently needed by both clinicians and engineers who work with or for wheelchair users. For instance, cadence and push length are important parameters that clinicians work on with newly injured wheelchair users, as they work effectively toward propulsion efficiency which, in turn, plays a very important role in secondary injuries occurred to wheelchair users. Also, tangential and total force applied on the rim is both effective on propulsion efficiency and determinative when measuring activities of wheelchair users. Moreover, starting angle is an effectual factor in adjusting wheelchair axle for stability of the wheelchair, and preventing secondary injuries. All of these show the importance of the factors reported in this paper as why they can be very informative.

One example is the average speed of wheelchair manoeuvring which found to be 0.85, about 57% to 77% of the average speed of SLP wheelchair propulsion reported elsewhere (references in the manuscript). When a factor like speed become nearly half in value, the meaningfulness of it is self-evident.

34 •Do the authors have any directions for future work based on these results? Some sentences were added on this subject.

 Conclusion: 

35 •This section should “wrap-up” the manuscript and highlight the results of this study. Consider adding a few more sentences to bring the manuscript to a succinct ending. Added.

Reviewer #2: 

36 It is inappropriate to use Null Hypothesis Significance Testing to argue for equivalence or equality. As it is typically used (i.e., when the researcher hopes to reject the null hypothesis), NHST is designed to make it difficult for the researcher to get what they want. When the researcher is hoping to 'prove' the null, however, as is the case here, achieving the desired result using NHST is quite easy: just design a poor and/or underpowered study. Failing to reject the null hypothesis means either: 1) that the null is true (in the case of this study, the repeated measurements are equal), or 2) the null is false (the repeated measurements are not equal) but the experiment doesn't have enough power to reject it. In order to arrive at the first conclusion (equality) the authors must provide evidence against the second (an underpowered study). Unfortunately, the authors provide no evidence of high statistical power. In fact, using only 11 subjects in the study makes it likely that the study is underpowered. In order for the results of the study to be meaningful, I strongly suggest that the authors forgo the use of NHST-based procedures such as MANOVA to determine equivalence in favor of more appropriate methods such as Bayesian methods, confidence intervals (Loftus, 1996; Aberson, 2002), or noninferiority testing (Streiner, 2003 The reviewer is correct in pointing out that NHST is not a suitable way to prove similarity. We agree and respectfully argue that the main evidence we are presenting to show the similarity here is the ICC between 16 variables on 32 measurement points, which is a standard statistical approach for proving reliability. The MANOVA analyses that the reviewer is pointing to, however, has been used to only confirm the proven idea by adding extra supporting material, and we made sure that we are very careful in the terminology we used for that in the manuscript to be clearer. For the MANOVA results, we just concluded that they were not different, not that they were the same. Again, the point raised by the reviewer is quite correct, but we need to clear up that this was not used as the main analysis in this study; only as supporting information.

37 If the authors insist on using NHST to claim evidence of equality, I view it as critical to demonstrate that the study as ran had high power. The sample size calculation based on 80% research power is added to the manuscript. 

38 Effect size measures, F statistics, and their associated degrees of freedom should be included for all MANOVAs. Sure. They are added to the manuscript now.

---

## [Decision Letter · Decision Letter 1]

17 Aug 2020

PONE-D-20-08803R1

Investigating the test-retest reliability of Illinois Agility Test for wheelchair users

PLOS ONE

Dear Dr. Salimi,

Thank you for submitting your manuscript to PLOS ONE. After careful consideration, we feel that it has merit but does not fully meet PLOS ONE’s publication criteria as it currently stands. Therefore, we invite you to submit a revised version of the manuscript that addresses the points raised during the review process.

We look forward to receiving your revised manuscript.

Kind regards,

Yih-Kuen Jan, PhD

Academic Editor

PLOS ONE

Reviewers' comments:

Reviewer's Responses to Questions

**Comments to the Author**

1. If the authors have adequately addressed your comments raised in a previous round of review and you feel that this manuscript is now acceptable for publication, you may indicate that here to bypass the “Comments to the Author” section, enter your conflict of interest statement in the “Confidential to Editor” section, and submit your "Accept" recommendation.

Reviewer #1: (No Response)

Reviewer #2: (No Response)

2. Is the manuscript technically sound, and do the data support the conclusions?

Reviewer #1: Partly

Reviewer #2: Yes

3. Has the statistical analysis been performed appropriately and rigorously? 

Reviewer #1: I Don't Know

Reviewer #2: Yes

4. Have the authors made all data underlying the findings in their manuscript fully available?

Reviewer #1: Yes

Reviewer #2: Yes

5. Is the manuscript presented in an intelligible fashion and written in standard English?

Reviewer #1: Yes

Reviewer #2: Yes

6. Review Comments to the Author

Reviewer #1: General Comments:

• The reviewer thanks the authors for their diligent revisions. The manuscript flows well with a clear direction.

• Minor grammatical errors throughout.

• Concerns remain for MANOVA statistical testing and interpretation.

• Further discussion is needed to address how results obtained from able-bodied individuals in wheelchairs are reasonably translatable to wheelchair users with disabilities.

Introduction:

• Lines 94-96: Minor wording. Consider moving “wheelchair maneuvering” before “training athlete wheelchair users” as maneuvering is the primary focus of the introduction and manuscript as a whole.

Methods:

• Lines 115 and 116: Using “average” in demographics is unnecessary after the authors indicate data listed are mean ± SD.

• Line 127: Consider using mean ± SD for consistency.

• Line 142: Justification is needed for why participants performed an agility test at a comfortable speed (rather than done as quickly as possible [Line 134])

Results:

• The ICCs shows good to excellent reliability, which is the main outcome of the study. If ICCs are high for most variables, this should be the sole justification for reliability. Concerns remain for the interpretation of MANOVA tests. Why supplement statistical testing with MANOVAs that fail to reject the null hypothesis? This would also eliminate any verbiage needed regarding Bonferroni corrections.

Discussion:

• Lines 380-386: References are needed for these statements.

• Future research is likely needed with non-able-bodied wheelchair users (i.e., participants with disabilities) to ensure repeatability of the IAT, especially if authors suggest that IAT can be used for future wheelchair maneuvering research.

Reviewer #2: Thank you for addressing my comments. I have no problems with the statistical procedures as they are presented in the revision, with one very minor exception. By my calculations, the power analysis conducted on page 5 suggests a sample size of 11.69 to achieve power of 0.8, so technically you should have used 12 subjects to achieve 80% power, not 11. As I said, though, I consider this a minor point.

As an aside, a few grammatical errors remain in the manuscript.

7. PLOS authors have the option to publish the peer review history of their article (what does this mean?). If published, this will include your full peer review and any attached files.

Reviewer #1: No

Reviewer #2: No

---

## [Author Response · Author response to Decision Letter 1]

5 Sep 2020

Comments Responses:

Reviewer #1: 

 Introduction: 

1 Lines 94-96: Minor wording. Consider moving “wheelchair maneuvering” before “training athlete wheelchair users” as maneuvering is the primary focus of the introduction and manuscript as a whole.

A: Done.

 Methods: 

2 Lines 115 and 116: Using “average” in demographics is unnecessary after the authors indicate data listed are mean ± SD. 

A: Very right. Corrected.

3 Line 127: Consider using mean ± SD for consistency. 

A: Sure. Corrected.

4 Justification is needed for why participants performed an agility test at a comfortable speed (rather than done as quickly as possible [Line 134]) 

A: Since completion time is not a great concern for wheelchair users when accomplishing wheelchair manoeuvres. Rather, other factors like how well they can finish the task without overusing their shoulders, are of more concern to them. An explanation is added to the manuscript [lines 141-142].

 Results: 

5 The ICCs shows good to excellent reliability, which is the main outcome of the study. If ICCs are high for most variables, this should be the sole justification for reliability. Concerns remain for the interpretation of MANOVA tests. Why supplement statistical testing with MANOVAs that fail to reject the null hypothesis? This would also eliminate any verbiage needed regarding Bonferroni corrections. 

A: As the reviewer pointed out, the main evidence we are presenting to show the similarity here is the ICC between 16 variables, which is a standard statistical approach for proving reliability. The MANOVA analyses, however, have been used to only confirm the proven idea by adding extra supporting material. Fortunately, the MANOVA results for ALL 37 variables were favorable and backed the main hypothesis of the study. From the MANOVA results, we concluded that the parameters were not different. We need to emphasize that “not rejecting the null hypothesis” was favorable for our study. This should clear up the misunderstanding and show that using MANOVAs not only did not weaken but also strengthen the results of the study.

 Discussion: 

6 Lines 380-386: References are needed for these statements. 

A: Six references were added.

7 Future research is likely needed with non-able-bodied wheelchair users (i.e., participants with disabilities) to ensure repeatability of the IAT, especially if authors suggest that IAT can be used for future wheelchair maneuvering research. 

A: A paragraph is added regarding this issue on lines 371 to 376.

Reviewer #2: 

8 I have no problems with the statistical procedures as they are presented in the revision, with one very minor exception. By my calculations, the power analysis conducted on page 5 suggests a sample size of 11.69 to achieve power of 0.8, so technically you should have used 12 subjects to achieve 80% power, not 11. As I said, though, I consider this a minor point. 

A: You are right. Rounding it to 12 was a better practice. For greater clarity, I denoted in the manuscript that the exact result of the formula was 11.63 (according to this:

1.5+((1.64+.84)/((ln((1+.6)/(1-.6))/2)-(ln((1+.9)/(1-.9))/2)))^2=11.63).

9 A few grammatical errors remain in the manuscript. 

A: We implemented another round of grammatical check and also applied the grammatical points raised by the first reviewer.

Editor: 

10 Upload your figure files to the Preflight Analysis and Conversion Engine (PACE) digital diagnostic tool 

A: Done.

---

## [Decision Letter · Decision Letter 2]

15 Oct 2020

Investigating the test-retest reliability of Illinois Agility Test for wheelchair users

PONE-D-20-08803R2

Dear Dr. Salimi,

We’re pleased to inform you that your manuscript has been judged scientifically suitable for publication and will be formally accepted for publication once it meets all outstanding technical requirements.

Kind regards,

Yih-Kuen Jan, PhD, University of Illinois at Urbana-Champaign

Academic Editor

PLOS ONE

Additional Editor Comments (optional):

Reviewers' comments:

Reviewer's Responses to Questions

**Comments to the Author**

1. If the authors have adequately addressed your comments raised in a previous round of review and you feel that this manuscript is now acceptable for publication, you may indicate that here to bypass the “Comments to the Author” section, enter your conflict of interest statement in the “Confidential to Editor” section, and submit your "Accept" recommendation.

Reviewer #1: All comments have been addressed

Reviewer #2: All comments have been addressed

2. Is the manuscript technically sound, and do the data support the conclusions?

Reviewer #1: Yes

Reviewer #2: (No Response)

3. Has the statistical analysis been performed appropriately and rigorously? 

Reviewer #1: Yes

Reviewer #2: (No Response)

4. Have the authors made all data underlying the findings in their manuscript fully available?

Reviewer #1: Yes

Reviewer #2: (No Response)

5. Is the manuscript presented in an intelligible fashion and written in standard English?

Reviewer #1: Yes

Reviewer #2: (No Response)

6. Review Comments to the Author

Reviewer #1: The reviewer would like to thank the authors for their diligent revisions. All sections of the manuscript are thorough and provide great detail. Few minor grammatical errors and alpha-numeric inconsistencies (Example: Line 325 - 1 vs. one) remain.

Reviewer #2: (No Response)

7. PLOS authors have the option to publish the peer review history of their article (what does this mean?). If published, this will include your full peer review and any attached files.

Reviewer #1: No

Reviewer #2: No

---

## [Editor Report · Acceptance letter]

19 Oct 2020

PONE-D-20-08803R2 

Investigating the test-retest reliability of Illinois Agility Test for wheelchair users 

Dear Dr. Salimi:

I'm pleased to inform you that your manuscript has been deemed suitable for publication in PLOS ONE. Congratulations! Your manuscript is now with our production department. 

Kind regards, 

on behalf of

Dr. Yih-Kuen Jan 

Academic Editor

PLOS ONE